# Overview of Repressive miRNA Regulation by Short Tandem Target Mimic (STTM): Applications and Impact on Plant Biology

**DOI:** 10.3390/plants12030669

**Published:** 2023-02-03

**Authors:** Syed Muhammad Iqbal Syed Othman, Arif Faisal Mustaffa, M. Hafiz Che-Othman, Abdul Fatah A. Samad, Hoe-Han Goh, Zamri Zainal, Ismanizan Ismail

**Affiliations:** 1Department of Biological Sciences and Biotechnology, Faculty of Science and Technology, Universiti Kebangsaan Malaysia (UKM), Bangi 43600, Selangor, Malaysia; 2Department of Biosciences, Faculty of Science, Universiti Teknologi Malaysia, Skudai, Johor Bahru 81310, Johor, Malaysia; 3Institute of Systems Biology, Universiti Kebangsaan Malaysia (UKM), Bangi 43600, Selangor, Malaysia

**Keywords:** microRNA, miRNA decoy, non-coding RNA, plant stress, plant development, short tandem target mimic, spatial-temporal promoter, target mimic

## Abstract

The application of miRNA mimic technology for silencing mature miRNA began in 2007. This technique originated from the discovery of the *INDUCED BY PHOSPHATE STARVATION 1* (*IPS1*) gene, which was found to be a competitive mimic that prevents the cleavage of the targeted mRNA by miRNA inhibition at the post-transcriptional level. To date, various studies have been conducted to understand the molecular mimic mechanism and to improve the efficiency of this technology. As a result, several mimic tools have been developed: target mimicry (TM), short tandem target mimic (STTM), and molecular sponges (SPs). STTM is the most-developed tool due to its stability and effectiveness in decoying miRNA. This review discusses the application of STTM technology on the loss-of-function studies of miRNA and members from diverse plant species. A modified STTM approach for studying the function of miRNA with spatial–temporal expression under the control of specific promoters is further explored. STTM technology will enhance our understanding of the miRNA activity in plant-tissue-specific development and stress responses for applications in improving plant traits via miRNA regulation.

## 1. Introduction

The global crop output is perpetually at risk due to ongoing climate change [1,2,3], with the impact of climate change expected to increase in the future [4]. Abiotic stresses resulting from climate change affect various physiological processes in plants, such as increasing transpiration rate, reducing carbon uptake, and decreasing respiration efficiency, which is caused by an interruption of the stomatal mechanism. These changes ultimately decrease crop productivity [5]. Despite these challenges, there is a need to increase agricultural yields by 70% in the next 30 years to support the global population, which is predicted to double by 2050 [6]. Thus, there is a dire need to explore and utilize various approaches to understand gene regulation and manipulation to ensure better plant performance and productivity [7].

The regulatory gene-expression network in plants involves several layers of regulatory components that control the biogenesis of genes, including signal transduction, chromatin remodeling, transcription factors, transcription, post-transcription, and translation [8,9,10,11,12,13]. MicroRNA (miRNA) is one of the critical components in regulating gene expression at the post-transcriptional level, playing essential roles in various molecular and developmental processes [14]. miRNAs are short, non-coding RNAs, approximately 19–24 nucleotides long. They confer gene-silencing abilities by cleaving messenger RNA (mRNA) and restricting the translation of transcripts [15,16]. miRNA has been found to play essential roles in various biological processes, including plant growth and development as well as biotic and abiotic stress responses [17,18,19,20]. Many miRNAs have been found to influence plant performance and yield-related agronomic traits, making them attractive targets for crop improvement [21,22].

Several approaches have been developed to help us understand the functions of miRNA in regulating gene expression, including the overexpression of *MIRNA* (*MIR*) genes, artificial miRNA (amiRNA), anti-microRNA oligonucleotides (AMOs), RNA interference (RNAi), transcription-activator-like effector nucleases (TALEN), clustered regularly interspaced short palindromic repeats/CRISPR-associated nuclease 9 (CRISPR/Cas9), and target mimics [23,24,25,26,27,28]. Traditionally, the overexpression and knockdown/knockout techniques are used in the functional investigations of miRNAs, which result in a gain-of-function (GoF) or loss-of-function (LoF), respectively. However, miRNA genetic mutants are less effective as their miRNAs are small in size and have numerous members with overlapping functions that spread over the intergenic regions [29]. Furthermore, miRNA overexpression does not completely demonstrate its role, as miRNA can regulate gene expression when it is upregulated or downregulated [30]. Since the discovery of the *INDUCED BY PHOSPHATE STARVATION 1* (*IPS1*) gene that downregulates miR399 activity in plants, a new way to inhibit miRNA using a mimicking target transcript strategy has been invented [31]. This review discusses the development of miRNA downregulating tools using mimicking techniques. We focus on the short tandem target mimic (STTM) as a reliable tool for studying the function of miRNA, including its potential to regulate specific plant traits in a tissue-specific and inducible manner.

## 2. miRNA Biogenesis and Its Regulation in Plant Transcripts

Initially, *MIR* genes are transcribed from gene promoters by RNA polymerase II, producing single-stranded RNAs that form a coiled-hairpin secondary structure due to near-perfect complementary repeat sequences. This process produces RNA duplexes known as primary transcripts (pri-miRNAs) [32]. The length and structure of these pri-miRNAs vary between miRNAs and typically range between 100 and 400 nucleotides [33,34]. These synthesized pri-miRNAs are then stabilized by the RNA-binding protein Dawdle (DDL). The conversion of pri-miRNA to the precursor miRNA (pre-miRNA) is carried out in plants by the nuclease-cleaving protein Dicer-Like 1 (DCL1) [35]. This conversion is conducted with the joint action of double-stranded RNA-binding protein Hyponastic Leaves 1 (HYL1) and the zinc-finger protein Serrate (SE) [36,37]. The DCL1, HYL1, and SE proteins can form dicing complexes and process pri-miRNAs into pre-miRNAs in dicing bodies (D-bodies) [38,39,40].

After the conversion of pri-miRNA into pre-miRNA, a miRNA/miRNA* duplex consisting of the guide strand (miRNA) and the passenger strand (miRNA*) will be produced through the action of DCL1. DCL1 cleaves the pre-miRNA, which is then methylated at the 3′ ends by sRNA methyltransferase Hua Enhancer 1 (HEN1) to protect it from exonuclease-mediated degradation [32,41]. The methylated miRNA/miRNA* duplex is then transported into the cytoplasm by the plant homolog of cytoplasmic exportin-5 protein, Hasty (HST) [42]. The miRNA duplex will be loaded into Argonaute 1 (AGO1) to form the RNA-induced silencing complex (RISC). The AGO protein is considered the most crucial structural protein in the RISC complex. It has four domains: namely, the N-terminal domain (N), the PIWI/Argonaute/Zwille (PAZ) domain, the MID domain, and the P-element-induced wimpy testis (PIWI) domain. The PAZ domain binds RNA, while the PIWI domain possesses RNase H activity [14]. The formation then turns the guide miRNA into a single strand while the passenger miRNA* is degraded. After that, miRNA-RISC carries out the mRNA-silencing interaction through translation inhibition or mRNA cleavage [38]. Subsequently, the RISC complex with guide miRNA (3′–5′) will recognize and bind to the target mRNA by complementary base pairing [43].

In plants, miRNAs usually show sequences that are almost perfectly complementary to target transcripts. Many miRNAs facilitate RISCs to target mRNA by the direct cleavage of mRNA that strongly represses gene expression. In addition, this process of repression can also be achieved by interfering with mRNA after its assembly with ribosomes in the cytosol or endoplasmic reticulum, a process called translational inhibition [44,45]. The miRNA-RISC cleaving targets usually depend on the complementarity of the 10–11th nucleotides from the 5′ end of the miRNA. Complementary-sequence base pairing in this region will cleave the mRNA, while non-complementary bases pairing sequences will inhibit mRNA at translation stages [14]. However, the findings on how miRNAs affect plant phenotypes are still not fully understood, as individual miRNAs can regulate the expression of many genes under different conditions [46].

## 3. Inhibition Mechanism of Target Mimic That Confers Mature miRNA Repression in Plants

Many approaches have been developed to better understand the specific role of miRNAs in regulating plant processes. The expression of miRNA genes usually depends on exogenous or endogenous stimuli toward the plant, while miRNA is also one of the essential components of plant regulatory networks. The roles of specific miRNAs can be studied either through miRNA GoF or LoF. To elucidate the miRNA GoF, overexpression is one of the approaches used to enhance the activity of the miRNA of interest. In this case, amiRNA overexpression is designed to increase the abundance of target miRNA in the cells. It is also possible to overexpress either precursor miRNAs (pre-miRNAs) or the cDNA of *MIR* genes identified from transcriptome sequencing [47,48]. Moreover, a vector that overexpresses miRNA in *Arabidopsis thaliana* and carries a constitutive promoter can effectively increase the expression of internal miRNAs. This method amplifies naturally occurring miRNAs using a primer designed and cloned into a vector [49]. However, the overexpression of miRNA alone may not accurately reflect the actual function of the miRNA, and it may result in undesirable plant performance. In *Chrysanthemum indicum*, the overexpression of *cin*-miR399a led to reduced salt and drought tolerance [48].

Thus, various LoF approaches have been developed to downregulate miRNA activity, such as targeting the *MIR* gene, pri/pre-miRNA, and mature miRNA [50,51,52]. However, to fully understand the function of individual miRNAs, it is important to investigate the role of mature miRNA activity in suppressing specific target transcripts. The LoF of specific, mature miRNA activities can be achieved by mimicking the mRNA transcript and sequestering the mature miRNAs from mRNA [53]. The miRNA mimic has been used to block miRNA activity by imitating the target transcript sequence. It also provides a mechanism that is resistant to cleavage by miRNA and degrades transcripts complementary to it at the 10–11th nucleotides [31,54]. The miRNA mimic also has a complementary sequence of the mature miRNA that will bind to the miRNA, avoiding the binding of the miRNA to the actual target mRNA. The miRNA mimic will protect the target transcript from suppression by acting as a decoy for the miRNA. This will increase the expression of target transcripts by reducing miRNA inhibition activities toward the target transcripts. This technology, known as a “target mimic,” was developed with exogenous and endogenous approaches to downregulating miRNA. An exogenous method is also known as anti-microRNA oligonucleotides (AMOs) (Figure 1a). Typically, AMOs rely on complementary base pairing between the oligonucleotide sequence and its target miRNA [55]. AMOs are chemically modified antisense oligonucleotides that have a sucrose-mediated delivery to plants. The inhibition of miRNA function can be achieved by immersing plant tissue in solutions containing AMOs and sucrose, with a more severe effect observed when the plant is treated with high concentrations of sucrose [56]. Previous research has demonstrated that AMOs have the ability to suppress the expression of particular miRNAs (miR156a–j, miR820a–c, and miR169f,g) in leaves. Depending on the sucrose concentration of the AMO solution, these downregulations display three distinct levels of expression: high, medium, and low. AMOs have also been observed to result in phenotypic effects when specific miRNAs (miR160a–d, miR167d–h, miR171g and miR390) are downregulated, including reduced root length and decreased root branching. Additionally, the application of AMOs can suppress miRNA in members of the same family without affecting members of other families. This offers a tool to research miRNA functions in a sequence-specific way, as well as to investigate the phenotypic effects mediated by miRNAs [56].

Other than these exogenous methods, several endogenous target mimic approaches have also been developed to mimic miRNA targets. These approaches utilize long, non-coding RNA within the plant genome that is complementary to the miRNA and functions through the *IPS1* mechanism of miRNA silencing, also known as the competing endogenous RNAs (ceRNAs) in mammals [31,57,58]. In addition, these endogenous, target-mimic transcripts also have properties similar to other non-coding RNAs (ncRNAs), which do not encode proteins but play a role in regulating the transcripts involved in developmental processes or stress responses [59]. However, endogenous miRNA target mimics intend to regulate miRNA activities instead of the gene transcript expression inside the cell. The discovery of the miR399-inhibiting non-protein-coding gene *IPS1* contributed to this finding in 2007. The *IPS1* and *PHOSPHATE 2* (*PHO2*) genes are involved in inorganic P (Pi) homeostasis, sharing a 23-nucleotide binding site that competes with miR399 for pairing [31]. However, *IPS1* contains three mismatch nucleotides that form a “central bulge” opposite the miRNA cleavage site, preventing it from being cleaved by miR399 activity [60]. This finding has led to the development and commercialization of miRNA-decoying techniques. Target mimicry (TM) is one of the first developed mimic tools, an artificial construct design based on *IPS1* (Figure 1b). Like *IPS1*, it has one binding site, but it has been designed with an amiRNA binding-site sequence for different decoy miRNAs. When constructed as an endogenous miRNA target mimic, it is known as MIMICS (MIM). Its mechanism is similar to *IPS1*, representing an miRNA target transcript and pairing with miRNA without the TM being degraded [31,61].

In addition to endogenous target mimics, RNA sponges—also known as molecular sponges (SP)—have been found in animals and plants that sequester miRNA, resulting in LoF (Figure 1c) [62]. Circular RNA (circRNA) was first observed in human and mouse brains. Its mechanism acts as a sponge by mimicking the miR671 target transcript, but it can be cleaved by miRNA [63]. However, with modifications, SP technology can have up to fifteen miRNA binding sites, including two mismatches that provide cleavage resistance to miRNA and four nucleotide spacers between the miRNA binding sites [64,65]. However, mimicking technology has demonstrated various levels of structural stability when transcribed, efficiency in decoying miRNA families or their members, and miRNA degradation [65].

To address this variability, A mimicking tool called STTM was developed (Figure 1d). STTM is an artificial, non-coding RNA technology that is expressed through genetic engineering techniques. It is a short RNA of less than 100 nucleotides. This tool is now the most well-understood and well-developed miRNA mimic. Furthermore, this technology was designed to be more stable in cells. It has more binding sites to decoy highly abundant miRNA. Thus, STTM was developed with modifications to the TM approach to increase effectiveness. STTM was designed with the most extended nucleotide spacer and a weak stem–loop structure, increasing its stability. In contrast, TM has no spacer, and SP only has a short spacer. STTM consists of two miRNA binding sites that can decoy different miRNAs, whereas TM only has one binding site, which may lower its effectiveness in highly abundant miRNAs [28]. In comparison, the SP targeting of certain miRNAs is complex to design and lacks a complete understanding of its mechanism, unlike STTM, which is easier to construct. Until recently, STTM has demonstrated its suitability to be studied with different miRNAs in model and crop plants, especially the conserved miR165/166 and miR156/157 with combined strategies [66].

## 4. STTM Application in miRNA LoF Studies

STTM was developed based on the concept of TM, which has constantly been evolving to better assist in the exploration of the role of miRNAs in plants and mammals. Among various artificial target mimics, STTM has been found to be the most effective and stable molecular mimic for regulating miRNAs in *Arabidopsis* [67]. Previously, the constitutive cauliflower mosaic virus (CaMV) promoter was used to develop both the STTM and TM constructs to observe the severity of the phenotype caused by miRNA silencing [61]. This STTM construct was compared with conventional, transgenic TM to observe the effect on the phenotype; it was found that the STTM induced a greater downregulation of miR166 [67]. This STTM was designed with several components that play an essential role in ensuring that the targeted miRNA is successfully silenced, such as small RNA binding sites, a spacer, three-nucleotide bulges, and the utilization of different promoters [52,66,68]. Thus, the development of STTM contributes to its application in targeting stress-triggered miRNA and other important crops (Figure 2) [66,68,69].

The spacer component serves to stabilize the STTM transcripts. This region consists of 48 to 88 nucleotides that form a secondary structure similar to a hairpin loop. Intermediate to this space is the *SwaI* site, which is used for the gene construction [52]. This spacer must have at least 48 nucleotides to ensure the stability of the construct and its effectiveness. Shorter spacers, such as those with 31 nucleotides, have been used, but decreased efficacy and stability have been observed when they are compared to spacers with more nucleotide length. However, the nucleotide length can be increased up to 96 nucleotides, showing a more significant phenotype towards silencing the miR165 and miR166 activities [28]. In addition, mutation on the nucleotide sequence of the spacer may disturb the formation of the STTM’s secondary structure, thus lowering the effectiveness of the STTM in miRNA suppression and STTM stability in cells [28,70]. Therefore, the spacer component has a minimum length of 48 nucleotides, and no mutation of the secondary structure is necessary to ensure the stability of the STTM construct to avoid self-binding and maintain the accessibility of the miRNA binding site [28,68].

Furthermore, STTM consists of two small RNA binding sites. These sites can use two different mimic sequences for targeting miRNA because they are separated by a spacer that forms a weak hairpin loop. These small RNA binding sites consist of 21 nucleotide sequences that are complementary to the mature miRNA. This miRNA binding site sequence masks or mimics the target transcript that is complementary to the mature miRNA [52,68]. A loop sequence is inserted around the 10th and 11th positions of the mature miRNA binding sites, causing the binding site of the miRNA complex to be resistant to cleavage, similar to the above-mentioned *IPS1* mechanism. The CUA and CAA sequences, as well as any other sequence that will produce a bulge loop when binding to miRNA, are frequently employed. However, these sequences must not exhibit any significant plant phenotype to optimize the severe phenotype of miRNA inhibition [68]. Additionally, these three nucleotides must not be complementary to the target miRNA at the 11th to 13th STTM sequences from the 5′ end in the middle or bulge sequence because miRNA can cleave other sequences that are complementary with the 10th to 11th nucleotides from the 5′ end of miRNA sequence [14,68]. Thus, this improvement in STTM constructs led to increased phenotypic alteration caused by miRNA inhibition when compared to previous STTM construct versions (Table 1).

## 5. Effectiveness of STTM in Regulating miRNA

STTM is a popular technology as it can target various miRNAs. It is well-established that miRNAs are transcribed from a wide range of non-coding genomic regions. The miRNA biogenesis pathway is a complex process that involves multiple enzymes and proteins, and mostly miRNA is transported into the cytoplasm for gene silencing (Figure 3a). Blocking one *MIR* gene transcript through gene deactivation by mutation methods will not be able to explain the entire functioning of the miRNA, as miRNAs have multiple members with overlapping functions over the intergenic region [71]. One of the highly conserved miRNA families in *Arabidopsis,* such as miR156, contains eight *MIR* genes that encode miR156a to miR156h. Thus, it is difficult to study its function through gene suppression [72]. STTM can inhibit miRNAs in various conditions by mimicking complementary to mature miRNA transcripts, inhibiting their gene-silencing activity (Figure 3b,c). In this approach, two short sequences mimic the target site of a small RNA separated by a linker nucleotide sequence and lead to the degradation of the small RNA to help understand the role of miRNA in regulating specific miRNA targets [28]. The degradation of miRNA by STTM involves the activity of Small RNA Degrading Nuclease (SDN). Although the mechanism of SDN in degrading targeted miRNA is still not fully known, a mutation towards SDNs (*SDN1-1* and *SDN2-1*) has shown no change in the abundance of target miRNAs and plant phenotypes that express STTM [28]. Additionally, the F-box protein Hawaiian Skirt (HWS) is triggered to degrade the miRNA that binds to the STTM mismatch loop which forms a non-optimal RISC in ubiquitin-dependent degradation [73]. The HWS has also been found to regulate the *IPS1* transcript at an upstream level that may correlate with the activation of STTM transcript biogenesis [74]. Despite this, there is still a gap in the complete comprehension of miRNA–STTM degradation involving SDNs and the HWS.

Additionally, four factors place STTM among the most effective tools for researching miRNA function in plants; these are supported by recent findings [28]. First, various miRNA families targeted by STTM have displayed a sharp and steady downregulation. Such miRNA suppression illustrates the potential of STTM for miRNA functional identification, including conserved miRNA families. However, the efficiency of STTM on different miRNA families can be affected by the three-nucleotide mismatch loop, which is designed to avoid the cleavage of STTM. Still, it can lower the free energy of binding if an excessive mismatch occurs [65]. Second, it appears that STTM and miRNA levels are inversely connected. When the STTM level increases, the targeted miRNA level decrease, and vice versa. For example, STTM interacted with miR165/166 quite effectively, reaching a 90% reduction of miR166 expression in *STTM166*-expressing plants, which usually led to the cleavage repression of *HOMEODOMAIN CONTAINING PROTEIN4* (*OsHB4*). Meanwhile, the abundance of miR160, which generally inhibits auxin response factor (ARF) genes such as *SlARF16A*, was decreased by 78% in *STTM160*-expressing plants via translational inhibition [28,75,76]. Thirdly, STTM only targets mature miRNAs and does not interfere with any pri/pre-miRNA, which lowers the likelihood of interfering with other processes in plants. STTM silencing also shows highly specific, desired phenotypes in monocotyledon and dicotyledon plants [69]. The fourth factor is that STTM targets miRNA from the 3′ end, indicating part of the mature miRNA’s mechanism in suppressing the mRNA target transcript. Mature miRNAs are complementary to the 5′ end of the targeted mRNAs. Therefore, an STTM mimicking the 5′ end sequence of the targeted mRNA is the most effective at silencing mature miRNA [28].

The effectiveness of the STTM approach has an advantage for various miRNA members and families. The miRNA in plants can be categorized into different members, each of which can be used to determine the relative abundance of a particular miRNA family member with different nucleotides. For example, the miR159 family has members a, b, and c, which differ in the last two nucleotides [70]. The conserved miRNA family contributes to more members such as miR169, which also can be manipulated by STTMs such as STTM169q and STTM169o, and can therefore downregulate specific miRNA members [77,78]. Therefore, a greater abundance of the member is adequate for the STTM study since it has two miRNA binding sites to comprehensively investigate the various targets of different members [79,80]. Furthermore, some members of the miRNA family have vastly different sequences, which prevents them from being mimicked by the same binding sites [79]. At the same time, STTM exhibits a high specificity without affecting the abundance of other miRNAs such as STTM156, which does not interrupt miR159 and miR160 abundance, making it suitable for targeting miRNAs from different families with specific binding sites [69].

## 6. Diverse Plant miRNA Functions Uncovered Using STTM

Although STTM has been applied to various plant species, its efficiency at miRNA inhibition is constantly being studied at the genetic and genomic levels, using *Arabidopsis* as a model plant [66,81]. *Arabidopsis* can be used to explore the effectiveness of a specific STTM before its application in a plant of interest to study gene functions or homolog interactions [82]. Based on the literature, the crops with the highest number of STTM applications are tomatoes (*Solanum lycopersicum* L.) and rice (*Oryza sativa*) (Figure 4). The tomato is a self-pollinated crop in the Solanaceae family with around 3000 species, and is the second highest in global vegetable production, after the potato [83]. Tomato plants become more resistant to powdery mildew if treated with drought, indicating plant resistance crosstalk to be more resistant toward other stresses [84]. In addition, STTM1916-transformed tomato plants are made more resistant to two bacterial infections, *Phytophthora infestans* and *Botrytis cinerea*, by upregulating strictosidine synthase (*STR-2*), UDP-glycosyltransferase (*UGT*), and MYB transcription factor *MYB12* [85].

Various phenotypes and traits expressed by model or crop plants can be modified by manipulating the miRNA activities using STTM. The STTM method has been widely used to examine the roles of various miRNAs across many different plant species, as is shown in Table 2. It was observed that the first STTM could influence the gene transcript of *PHABULOSA* (*PHB*), *PHAVOLUTA* (*PHV*), *REVOLUTA* (*REV*), and members of a small homeodomain-leucine zipper family, *ATHB8* and *ATHB15,* by inhibiting miR165/166 activities in *Arabidopsis* [28]. Subsequent studies showed that STTM could efficiently manipulate diverse families of miRNAs for the discovery of miRNA functions [52,69].

On the other hand, rice is a major staple food crop in most countries [130]. Therefore, it is desirable for rice performance to have the ability to tolerate different stresses as the most important characteristic in crop improvement for increased yield. STTM helped to reveal the interaction between rice miR172 and *INDETERMINATE SPIKELET1* (*IDS1*) in salt stress;miR319 improved yield through an increase in tiller bud and grain with upregulated *TEOSINTE BRANCHED1*/*CYCLOIDEA*/*PCF* (*OsTCP21*) and *GIBBERELLIN AND ABSCISIC ACID REGULATED myb* (*OsGAmyb*) [87,128]. These studies demonstrated the important applications of miRNA in improving phenotypes, especially in crops [22,131,132,133,134].

### 6.1. Plant Development and Architecture

STTM is an effective tool for functional genomics studies in plant development, especially plant morphology and growth throughout different developmental stages. STTM generates a clear phenotype in plant architecture to elucidate miRNA function in plant development. STTM-miR165/166, (STTM165/166)-transformed *Arabidopsis* with upregulated class III homeodomain/Leu zipper (HD-ZIP III) transcription factors (*PHB*, *PHV* and *REV*), showed an abnormal phenotype resulting from the loss of apical dominance and leaf asymmetry [28]. In addition, the miR165/166 copy number decreased to almost zero due to the inhibitory effect of STTM165/166 [52].

STTM has also been used to uncover miRNA’s involvement in stem cell formation. The biosynthesis of lignin in poplar plants could be manipulated using STTM to downregulate miR828, which directly targeted two MYB genes (*MYB171* and *MYB011*). This led to an increased lignin deposition by activating *PHENYLALANINE AMMONIA LYASE 1* (*PAL1*) and *CINNAMOYL-COA REDUCTASE2* (*CCR2*), both of which were highly abundant in the stem during secondary vascular formation [116]. STTM also increased the number of axillary branches of a tomato plant by silencing sly-miR171 [118]. The functional genomics of root miRNA revealed that miR160a/b in potatoes affected specific tissue regulations of transcripts. Potato plants with STTM mimicking the target for miR160a/b showed shorter lateral roots and an increased lateral root number, but also demonstrated a decreased root weight [80].

STTM319 targeting miR319 in wheat increased the transcripts of TCP (*TaPCF8*) and GAMYB (*TaGAMYB3*) transcription factors, affecting plant heights, tiller numbers, spike leaves, and wheat-grain yield [115]. STTM319 is also effective in rice to obtain a higher number of tiller buds and a greater yield despite different species [128]. In soybeans, GmSTTM166 demonstrated the function of miR166 in regulating plant height by targeting the *ATHB14-LIKE* that directly represses the expression of the gibberellin biosynthesis gene. GmSTTM166 silenced miR166, which caused the plant to dwarf because of an upregulated *ATHB14-LIKE* transcript [120]. A correlation between miR166b and the START domain-containing protein gene (*OsHox32*) in determining the rice plant structure has also been shown. STTM166b demonstrated the downregulation of the miR166b interaction with *OsHox32*, which affected the overexpression of *OsHox32* (OE*Hox32*). OE*Hox32* plants have shorter internodes and a reduced cellulose content and lignin thickness, similar to STTM166b-transformed plants. The lack of cellulose and lignin makes the leaf droop and easily break when bending, showing less strength than wild-type or RNAi-*OsHox32* plants. However, STTM166b did not interrupt rice plant height like in soybeans [113].

In the poplar plant, the STTM393 transformant had ten times less miR393 than the controls. After three months, the STTM393 plants grew 20% taller, 15% thicker, and had two to four more numbers of internodes than the control plants. Cross-sections of the STTM393 transgenic plant stems showed broader phloem, xylem, and cambium cell layers than the control plants. Additionally, their lignin content was higher than in non-transgenic plants [119]. In addition, STTM393 plants displayed an increased expression of an auxin signaling pathway, cell cyclin, cell expansion, and lignin production genes. Higher *FBL* expression levels showed that the STTM393 plants had a more active auxin signaling pathway to stimulate plant development [119]. In switchgrass, miR396 was found to target the growth-regulating factor (GRF) gene module (*PvGRF1*, *3*, and *9*) involved in regulating plant height and lignin content. The overexpression of miR396 (OE*MIR396*) produced plants with reduced height, internode length, and stem dry biomass. Thus, miR396 downregulation could maintain plant architecture and strength [135].

### 6.2. Leaf Development

Leaf morphology and physiology can significantly influence photosynthetic performance. The total area of the leaf, xylem, and phloem influence the rate of photosynthesis [136]. Various STTM designs revealed the miRNA function in leaf development. Transgenic *Arabidopsis* with STTM169d showed an increased number of rosettes and leaf size with the upregulation of auxin response factors (*ARF1* and *ARF2*) [108]. STTM827 suppressed miR827 to less than 10% in regulating leaf senescence and phosphate homeostasis. STTM827 delayed leaf senescence in natural and dark-induced conditions by enhancing the expression of the GLABRA1 enhancer-binding protein (GeBP)-like (*GPLα*) that suppresses senescence and *NITROGEN LIMITATION ADAPTATION* (*NLA*), which is a suppressor for senescence transcriptional activator gene, *ORE1*. Meanwhile, phosphate homeostasis was reduced by the negative regulation of *GPLα* or *NLA* on *PHOSPHATE TRANSPORTER 1* (*PHT1*), which encodes for a plasma-membrane-localized phosphate transporter [109]. STTM can also be used to explore the function of leaf miRNA in both woody and rice plants. STTM-BpmiR164 was used in *Betula pendula*, leading to an increase in the expression of *CUP-SHAPED COTYLEDON 2* (*BpCUC2*), which was targeted by miR164, resulting in abnormal leaf shapes and shorter internodes. In rice, STTM364 suppressed the miR364 that targeted *LEAF INCLINATION 4* (*LC4*), leading to an elevation in leaf inclination [110,111]. The Overexpression of SlymiR208 in transgenic lines suppressed the cytokinin biosynthesis genes (*SlIPT2* and *SlIPT4*), which induced leaf senescence at an early stage. Thus, downregulating the abundance of SlymiR208 potentially reduces young-leaf senescence in tomatoes [137].

### 6.3. Root Development

The spatial layout of the root system (number or length of lateral organs) varies significantly based on plant species, soil composition, and especially the availability of water and mineral nutrients. It involves abiotic and biotic environmental cues [138]. The function of miRNA in root development, such as the root length, is revealed through the application of STTM. For example, silencing the miR396a, which regulates *BASIC HELIX-LOOP-HELIX TRANSCRIPTION FACTOR 74* (*bHLH74*), showed longer root systems in *Arabidopsis*. Additionally, miR156 was found to target *SQUAMOSA PROMOTER BINDING PROTEIN-LIKE* (*SPL*) that affects the root meristem by altering auxin and cytokinin responses [139]. *Arabidopsis* plants with low amounts of miR156 had a smaller meristem size, resulting in a shorter primary root (PR). On the other hand, it was found that the repression of miR390 by STTM390 negatively impacted the lateral root development and salinity tolerance in rice [123]. In poplar, the overexpression of miR167a that suppressed *PeARF8.1* enhanced lateral root growth, while the overexpression of *PeARF*s reduced lateral root development [140]. The STTM-mediated downregulation of miR167a increased the activity of *PeARF*s, resulting in increased adventitious root development.

### 6.4. Flower Development

The physiological function of a flower is to facilitate reproduction, which is often accomplished by the joining of sperm and eggs with fruit growth. Floral abnormalities, anther dehiscence inhibition, and low pollen fertility can influence this function. Flower development can be manipulated to enhance fruit production, especially under sub-optimal conditions. In citrus, CsmiR399a.1-STTM plants grown under conditions with an adequate supply of inorganic phosphate (Pi) showed a reduced total phosphorus content in their leaves due to the upregulation of *PHO2*, a ubiquitin-conjugating E2 enzyme (*UBC24*) that interacts with the floral regulator genes from the SEPALLATA family (*CsSEP1.1*, *CsSEP1.2*, and *CsSEP3*), and the anther dehiscence regulator *INDUCER OF CBF EXPRESSION1* (*CsICE1*), which mimics a plant Pi-deficiency condition. Thus, CsmiR399a.1-STTM plants displayed the typical symptoms of Pi deficit to plant floral development, such as aberrant floral development, suppression of anther dehiscence, and reduced pollen fertility [100].

STTM156 induced early flowering in transgenic cotton plants by upregulating the floral development gene. The miR156 targeted *SQUAMOSA-PROMOTER BINDING PROTEINLIKE* (*SPL*), which activated *LEAFY* (*LFY*), *FRUITFULL* (*FUL*), and *APETALA1* (*AP1*). Thus, the downregulation of miR156 silenced *SPL* and upregulated other genes for floral development [102]. Other miRNAs identified to be involved in flower development include miR172-*APETALA2* (*AP2*) in black goji. The downregulation of miR172 by STTM172 will upregulate its target *AP2*, a repressor for *FLOWERING LOCUS T* (*FT*). Subsequently, this activates the expression of the flowering integration factor (*SOC1*) that was inhibited by *FT*, thus induced an early flowering time [141].

### 6.5. Fruit Development

STTM has also been applied to demonstrate the modulation of fruit shape and size through the alteration of miRNA. STTM159-silencing of the sly-miR159 that targets *SIGAMYB2* produced larger and heavier tomato fruits. The role of sly-miR159 in fruit formation was also confirmed by the CRISPR/Cas9 mutation of pre-sly-miR159 [107]. Notably, sly-miR159 overexpression did not significantly affect gibberellin levels in tomato-fruit formation [107,142]. This study demonstrated the quantitative effect on organ development from the manipulation of a small, non-coding RNA.

Additionally, STTM is also useful in investigating the functional genomics of post-harvest. The knockdown of Sly-miR164a by STTM164a revealed that the hardness of tomato fruit increased with a lower content of hydrogen peroxide; while an increased ABA content conferred a higher tolerance to chilling harm during periods of low temperature [104]. Attempts have been made to identify miRNAs that affect fruit development in plants. In tomato plants, miR157 overexpression resulted in the downregulation of *COLOURLESS NON-RIPENING* (*CNR*), the genes that turn young, immature fruits into ripe fruits. Hence, miR157 downregulation could delay fruit ripening and improve fruit harvesting time [143]. Other findings on the correlation of miRNA with fruit development include miR393/miR160/miR167-*TRANSPORT INHIBITOR RESISTANT* (*TIR*)/*AUXIN SIGNALING F-BOX* (*AFB*)/*ARF* and the miR477-antisense long non-coding RNA, *ABCB19AS* [144,145].

### 6.6. Secondary Metabolite

STTM397 was used to study the function of DkmiR397 in the biosynthesis of proanthocyanins (PA) in persimmon fruits. DkmiR397 targets the laccase gene *DkLAC2*, which is a key enzyme in the biosynthesis of PA. The presence of higher levels of *DkLAC2* in persimmon plants transformed with STTM397 resulted in an increased PA accumulation [126]. In tomatoes, STTM858 showed an increased anthocyanin under normal conditions, whereas *SlMYB7*-like transcripts considerably increased when miR858 was downregulated. The concurrent increased expressions of several anthocyanin downstream biosynthesis genes, including *PHENYLALANINE AMMONIALYASE* (*PAL*), *CHALCONE SYNTHASE* (*CHS*), *DIHYDROFLAVONOL REDUCTASE* (*DFR*), *ANTHOCYANIDIN SYNTHASE* (*ANS*)*,* and *FLAVONOL-3-GLUCOSYLTRANSFERASE* (*3GT*), caused a heavy accumulation of anthocyanins in the leaves, stems, and leaf buds of transgenic plants [124]. Recently, miR172 overexpression showed a decreased red coloration and anthocyanin accumulation in apples and, similarly, in *Arabidopsis*. This was due to the downregulation of the *AP2* gene, which acts as a positive regulator for the anthocyanin biosynthesis [146].

### 6.7. Biotic and Abiotic Stresses

STTM has the potential to increase plant stress tolerance. The downregulation of miR164 by STTM164 conferred a tolerance of iron (Fe) deficiency by increasing the primary root length, the number of lateral roots, and the transcripts of *IRON-REGULATED TRANSPORTER 1* (*IRT1*) and *FERRIC REDUCTION OXIDASE 2* (*FRO2*) when the Fe nutrient was scarce in the soil [147]. STTM can also increase plant tolerance to high salinity by suppressing certain miRNAs [78,86]. STTM169q suppresses ZmmiR69q, which is normally downregulated during salinity stress to regulate reactive oxygen species (ROS) accumulation by activating *PEROXIDASE1 (POD1*) via the *NUCLEAR FACTOR Y SUBUNIT A 8* (*ZmNF-YA8*) module in maize [78]. The STTM169q-transformed plants showed a higher survival rate (40–47%) than the wild type, which was higher than the overexpression of the ZmmiR169q target gene *ZmNF-YA8* (24–45%). Meanwhile, the overexpression of ZmmiR169q resulted in low survival rates and smaller plants than the wild type. Another study reported that cin-miR396a was involved in the regulation of drought and salt stresses in chrysanthemum. The overexpression of cin-miR396a reduced the plant tolerance toward both abiotic stresses, causing decreased leaf water content and leaf-free proline content. Additionally, cin-miR396a targets the stress-induced growth regulatory factors (*CiGRF1* and *CiGRF5*) [48].

On the other hand, STTM can also help to improve biotic stress resistance. STTM482b and STTM482 have been demonstrated to confer plants with a greater resistance to pathogen attack [97,98]. *Phytophthora infestans* poses a major threat to tomato plants and interacts with miR482b, *NUCLEOTIDE-BINDING SITE* (*NBS*), and the disease-resistance gene *LEUCINE-RICH REPEATS* (*LRRs*). Overexpression of miR482b reduced the resistance of tomato plants toward *P. infestans* with darkened leaves and gradual cell death. STTM482b successfully reduced miR482b abundance and activated *NBS-LRR* genes to make the plants more resistant to this pathogen [98]. This was further confirmed using STTM482- and STTM2118b-targeting miR482 and miR2118b, respectively, upon *Pseudomonas syringae* and *P. infestans* infection in tomatoes; these are the regulators for the disease-resistance gene *NUCLEOTIDE-BINDING LEUCINE-RICH REPEAT* (*NLR*) [97]. There are various miR482 subfamilies, such as miR482e downregulation by STTM482e, which also enhanced tomato resistance against pathogen attack with the manipulation of *NBS-LRR;* this can be further explored using STTM [148]. There are still many unexplored miRNAs that could increase plant resistance to disease attack. In tomatoes, miR6024 targets the *NLR* gene in *Alternaria solani* necrotrophic disease infection. The overexpression of miR6024 resulted in plants showing more lesions, higher ROS generation, and a hypersusceptibility to *A. solani* than wild-type plants [149].

### 6.8. Yield

The agronomic characteristics of crop plants or yield can be measured through grain weight, the number of panicles produced by each plant, and the total amount of grains produced by each panicle [150]. Using STTM, rice agronomic traits regulated by miRNA can be manipulated. For example, the overexpression of STTM-miR398 in rice uncovers its function in rice-yield traits. The STTM-miR398 suppression of miR398 affected yield traits, such as decreasing the seed size, tiller number, and plant height, indicating the important role of miR398 in rice productivity [69]. Furthermore, *OsTCP21* and *OsGAmyb* overexpression lines with STTM319 increased the length and weight of tiller buds [128]. miR1432 showed a higher expression in inferior grain, i.e., a higher yield of superior grains with a lower expression of miR1432 during seed development [151,152]. The downregulation of miR1432by STTM1432 increased the overall grain output by increasing the grain filling rate and weight in rice [129]. This was achieved through the suppression of miR1432 in seeds, leading to an upregulation of the *ACYLCOA THIOESTERASE* (*OsACOT*) gene module, which plays a role in the metabolism of fatty acids and phytohormone biosynthesis, specifically auxin and abscisic acid.

On the other hand, photosynthesis also plays an important role in plant yield via biomass production. Photosynthesis is a complex pathway in plants that causes plants to be autotrophic. Generally, it consists of reactions involving light, water, and carbon dioxide that produce glucose and oxygen [153]. There has only been a limited amount of validation to prove the involvement of miRNA in the regulation of photosynthesis; however, a wide range of bioinformatic or computational techniques showed the potential of miRNA in regulating photosynthesis [154]. Studies have been performed in rice on its tolerance to low light and its photosynthesis. The osa-miR2102-targeting chlorophyll a-b binding protein (CAB), which is a part of the protein in the antenna complex, and an osa-miR530-targeting ubiquinone biosynthesis protein (COQ4) involved in electron transport, were downregulated in tolerance rice. These miRNAs, which are involved in low-light responses in tolerance rice, were upregulated in the sensitive plants, resulting in a constant chlorophyll a content and an increase in chlorophyll b content for a greater extent of capturing solar energy and maintaining the photosynthesis process. Hence, STTM could be applied to manipulate the downregulation of these miRNAs to enhance desired traits, such as the maintaining of photosynthesis or increasing photosynthesis efficiency through the upregulation of chlorophyll b content or electron transport efficiency [155].

## 7. Application of STTM as a Constitutive and Spatial–Temporal miRNA Repressor

Promoters, enhancers, and silencers are gene-regulatory elements. They are involved in activating or suppressing gene transcription, including instances involving an artificial gene. In nature, gene regulation plays a critical function in cell division, growth, and development, and the plant defense system. This regulatory system is triggered by endogenous or exogenous stimuli. *Cis*-regulating modules control gene transcription and can act as a promoter, enhancer, silencer, insulator, or multifunctional sequence element [156,157]. The promoter is a *cis*-regulating element that begins and regulates the transcription at the upstream site of an associated gene as a part of major-type *cis*-regulatory modules [158]. These promoter DNA sequences often are found in the 5′ region of a functioning gene. There are two crucial areas in a promoter: (1) the core area in which RNA polymerase II attaches and initiates transcription at the base level, and (2) proximal and the distal regions that harbor multiple *cis*-regulatory motifs for the spatial–temporal control of gene expression [159].

Furthermore, STTM is an effective tool for analyzing the LoF with negative-regulatory miRNA [61]. Most STTM research design uses constitutive promoters to gain severe phenotypes caused by inhibiting miRNA activities [68]. Constitutive promoters enhance the continuous and high expression of STTM in plants, such as *CaMV35S* (Figure 5a) [52]. These CaMV promoters have high activity under normal and stressful conditions in almost all tissues. The *AtSCPL30* promoter (PD7) is an alternative constitutive strong promoter to *CaMV35S* as it is an endogenous plant promoter [160]. Constitutive promoters reveal the miRNA function indiscriminately in any plant organs, developmental stages, or growth conditions. Furthermore, a *CaMV35S* promoter may not be active in certain organs or plant stages [161]. The activation of a gene in a specific tissue or organ (spatial) and in specific developmental stages or stimuli (temporal) can be achieved with the tissue-specific (Figure 5b) and inducible (Figure 5c) promoters [66,129]. Therefore, STTM applications can be manipulated with specific promoters. These different promoters allow for the spatial and temporal analysis of miRNA functions with the precise manipulation of desired crop attributes. Homologs of promoter and transcription factors could be useful for having the same functions in different plant species [162].

In a study by Peng et al., STTM167 and STTM1432 were constructed with Gt13a promoters to activate the gene in a seed-specific manner, specifically in the endosperm cell of rice [66]. The STTM1432 was used to suppress the OsmiR1432, leading to a high expression of *OsACOT* [129]. The downregulation of the miR1432-*OsACOT* module via STTM1432 resulted in a superior phenotype, including higher yield and disease resistance, when compared to the constitutive overexpression of *OsACOT* that only enhanced the yield with an unchanged blast disease resistance [163]. Many promoter sequences can be explored for tissue specificity, such as seed or fruit, flower, tuber, anther, root, and leaf, which can be utilized to express STTM in a particular organ [162,164,165,166,167,168].

Many genes can also be activated by the process of inducible expression, which is based on reactions by triggers of physical, chemical, or environmental stimuli as an external factor [169,170,171]. Furthermore, previous studies have shown that the expression of STTM can be regulated using inducible promoters, such as the β-estradiol promoter. For example, iSTTM165/166 is activated when treated with β-estradiol, allowing for the regulation of STTM transcript expression [66]. The activated construct with the application of β-estradiol to the plant showed a similar phenotype as the constitutive promoter. In addition, this phenotype was not caused by the β-estradiol chemical, as wild-type plants treated with β-estradiol did not show any different phenotype when compared to the untreated plants. It is also worth noting that the use of STTM constructs driven by inducible promoters allows for more precise control over the timing and duration of miRNA suppression, which can be useful in certain experimental contexts. Additionally, using an inducible promoter may help to minimize any potential off-target effects that may occur with the constitutive expression of the STTM. The phenotypes exhibited by both STTM165/166 and iSTTM165/166 were similar, including curling leaves and a loss of adaxial-abaxial polarity due to the high levels of anthocyanin. Furthermore, this inducible knockdown used pER8 and pCXGUS-P vectors, which is different from a common, constitutive STTM [171]. Most of these inducible promoters can help induce plant tolerance to abiotic stress, which is mostly concerned with water deficit and salt stress [172]. In another miRNA study, biotic stress promoters such as AGO18 were used, which activated the expression of construct when they were induced by viral infection [173].

## 8. Future Direction of STTM as Alternative Ways to Explore miRNA in Different Spatial–Temporal Expressions

STTM has been shown to be an effective method for investigating the function of miRNA at the molecular level or in the context of trait analysis. This technique can be used to examine different miRNA activities in other plant species at different growth phases. It can also show desirable features in various potential candidate miRNAs that regulate genes in specific tissues [14]. However, there is still a lack of data regarding miRNA roles in various plant species and plant adaptation to stresses that can be discovered through STTM application. The discovery of miRNA functions in regulating key genes may provide an additional perspective on how we may maintain our food security in the face of climate change.

The activity of miRNA can be manipulated in different ways, mainly through the overexpression and downregulation of miRNA [80]. An miRNA functional study in regulating gene expression using inducible or tissue-specific promoters opens up new ways to study and manipulate miRNA for desired traits in essential crops such as rice [66]. Understanding gene regulation in grain crops can improve disease resistance, drought or salinity tolerance, and grain yield. Using STTM, which regulates miRNA activity under suitable spatial and temporal conditions, provides ways to improve traits without interfering with normal phenotypes and reduces the off-target effects in inhibiting non-conserved miRNA. However, it is challenging to identify targeted miRNA activations as some are expressed differently in different organs and mainly by stress triggers [174,175]. There are known strategies to help in the identification and expression of known and novel miRNAs, such as miRNA profiling with the various techniques combined, including high-throughput sequencing technology and the analysis of differentially expressed genes (DEGs) [176,177]. MicroRNA profiling can facilitate a greater understanding of the regulation of miRNAs in various organs and their response to different stresses, as some miRNAs are only expressed in the presence of stressors [178,179]. Thus, understanding specific miRNA expressions will open up new strategies for STTM application to improve plant traits with desired gene expressions through the targeted downregulation of miRNA.

## 9. Conclusions

The study of miRNA function in plants has been aided by the development of various mimicking tools which can potentially improve plant traits. It is important to carefully consider the effectiveness and stability of these tools when choosing a strategy for decoying miRNA in post-transcription stages. STTM is a promising tool for suppressing miRNA expression in various plant species, tissues, and developmental stages. The versatility of STTM enables the exploration of the conserved and novel functions of miRNAs in model and crop plants. Many miRNAs regulate transcripts in specific plant tissues or different stress conditions. STTM technology can be customized to achieve expression in specific tissues or in response to specific stimuli by manipulating the STTM promoter region. This manipulation can help to reduce off-target effects. The expression of STTM at specific stages can be useful for elucidating miRNA function without confounding effects on plant yield or phenotype.

## Figures and Tables

**Figure 1 plants-12-00669-f001:**
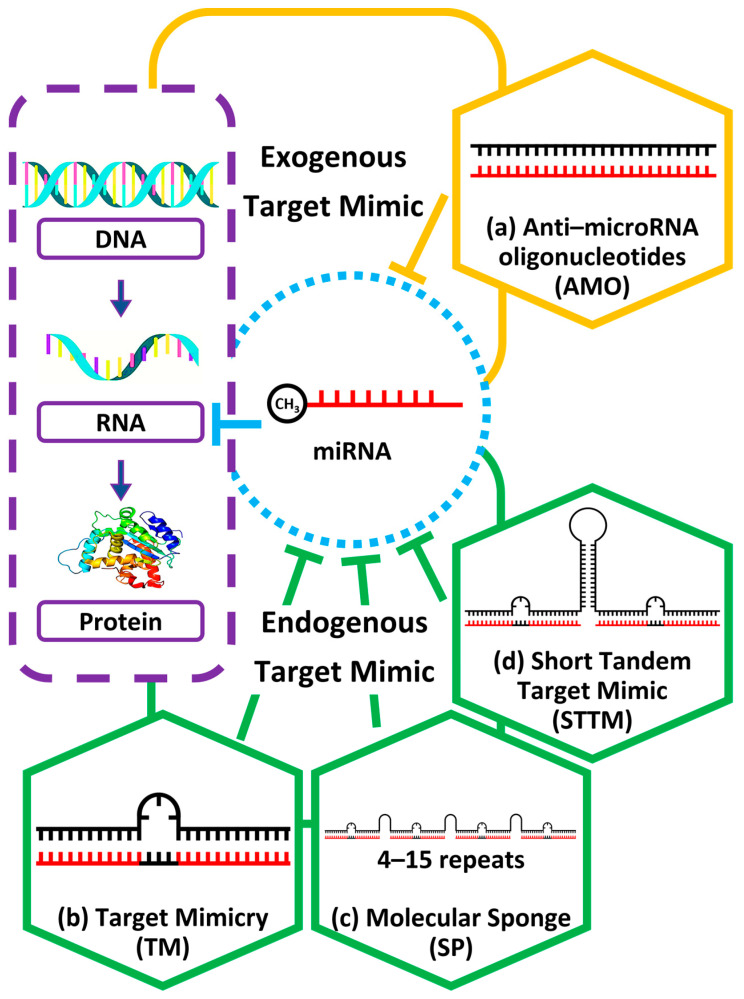
Exogenous mimicking approaches by using (**a**) anti–microRNA oligonucleotides (AMO) while endogenous approaches include (**b**) target mimicry (TM), (**c**) molecular sponges (SP) and (**d**) short tandem target mimic (STTM). The mechanism used to mimic the miRNA varies from its secondary structure and the number of miRNA binding sites.

**Figure 2 plants-12-00669-f002:**
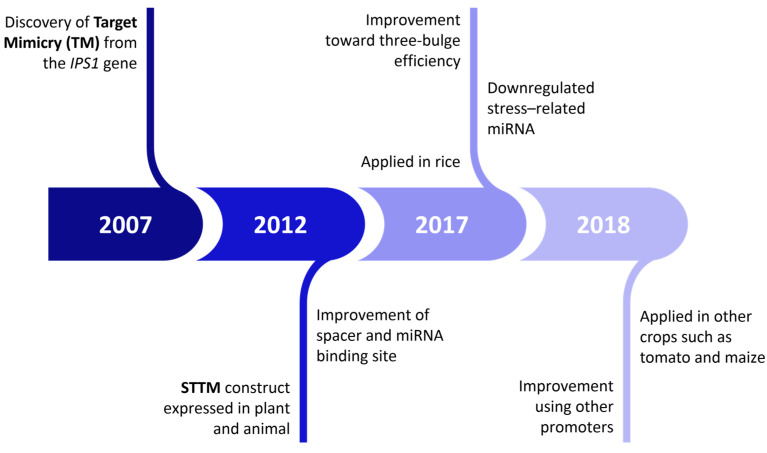
A historical timeline of the STTM approach developments that increased its efficiency and application in other plants.

**Figure 3 plants-12-00669-f003:**
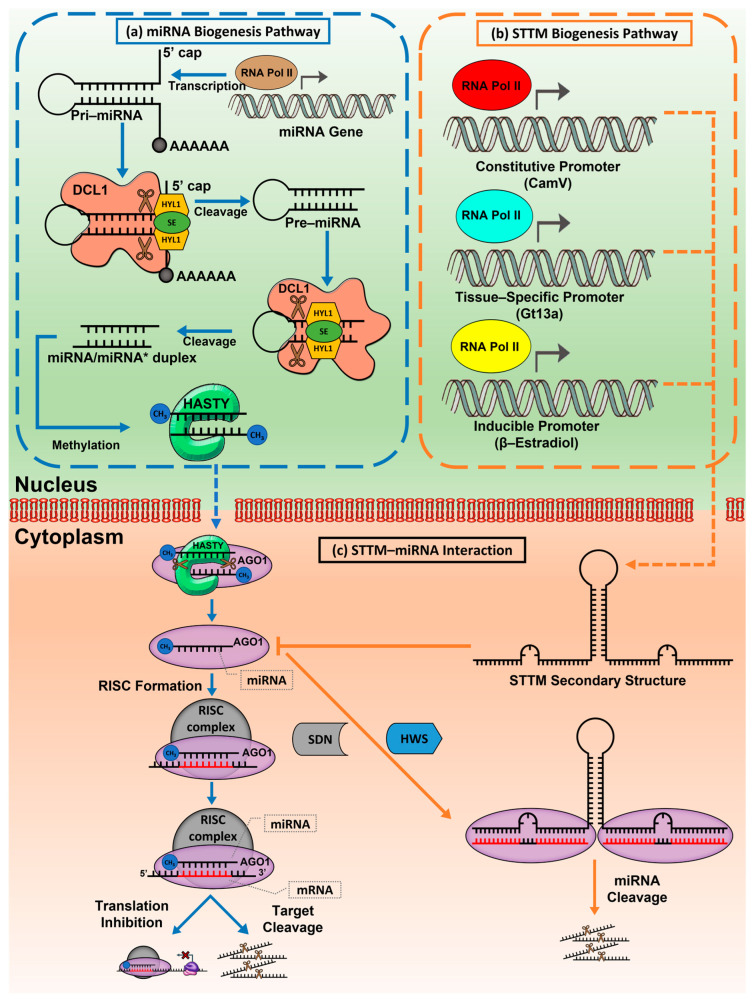
(**a**) miRNA biogenesis and (**b**) STTM expression during different conditions: constitutive expression, tissue-specific expression, and inducible expression in the nucleus while (**c**) in cytoplasm show the degradation process of miRNA–mRNA and the STTM interference in miRNA degradation that also inhibits its activity.

**Figure 4 plants-12-00669-f004:**
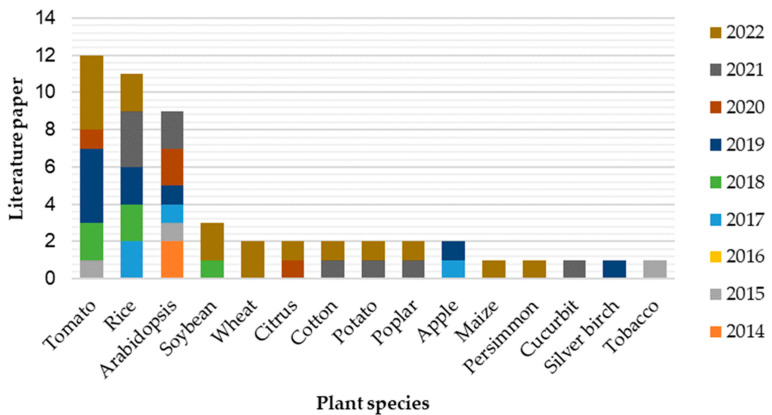
Reports on the STTM study of miRNA in different plant species since 2014. Tomato, rice, and *Arabidopsis* dominate the number of the publications.

**Figure 5 plants-12-00669-f005:**
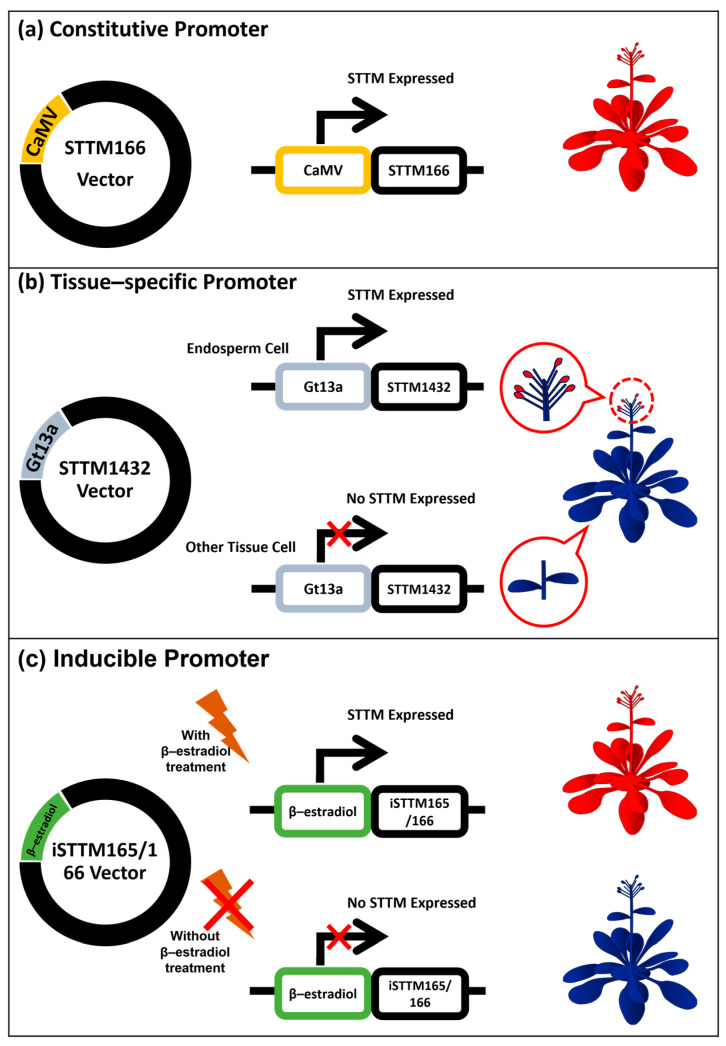
Activation of STTM expression in three different spatial–temporal types of miRNA silencing: (**a**) a constitutive promoter for continuous expression in cell to various tissues and conditions; (**b**) as a tissue-specific promoter expressing STTM in the endosperm cell but not in other tissues; and (**c**) as an inducible promoter that expressed STTM only when plants were treated by a chemical trigger (β-estradiol). The red color indicates STTM expression, while the blue color indicates no STTM expression.

**Table 1 plants-12-00669-t001:** Comparison of advancements in the development of STTM components and their correlation with changes in plant phenotypes.

Component	No/Weak Phenotypic Alteration	Strong Phenotypic Alteration
Spacer construct	Absence	Present
STTM expression	Low expression	High expression
Spacer lengths	Lengths of 8 and 31 nucleotides	Lengths of 48, 88 and 96 nucleotides
Stem region structure	Stem region disrupted via mutation, even though a long spacer was used	Stable stem region with no mutation
Number of miRNA binding sites	One binding site	Two binding sites
Mutation of miRNA binding sites	Mutation occurs	No mutation occurs
Tri-nucleotide bulge	Complementary to 10th to 11th	Not complementary to 10th to 11th

**Table 2 plants-12-00669-t002:** Diverse plant species and STTM targets for miRNA functional discovery.

Plant	STTM Target(s)	Target Gene(s)	Trait	Observed Effects	References
Rice	miR398	*CSD1–2, CCS*	Abiotic resistance	Tolerance to salinity but growth inhibited under normal conditions	[86]
Maize	ZmmiR169q	ZmNF-YA8	Abiotic resistance	Enhanced plant salt resistance	[78]
Rice	miR172	*IDS1*	Abiotic resistance	Reactive oxygen species (ROS) regulation	[87]
Wheat	miR164	*TaNAC14*	Abiotic resistance	Impacted root development and growth and stress (drought and salinity)	[88]
Apple	MdmiRln20	*Md-TN1-GLS*	Biotic resistance	Reduced Glomerella leaf spot (GLS) incidence	[89]
Apple	MdmiR156ab MdmiR395	*MdWRKYN1MdWRKY26*	Biotic resistance	*WRKY*-regulated pathogenesis-related (PR) protein-encoding genes boost plant biotic resistance	[90]
*Arabidopsis*	miR825/825	*AT5G40910* *AT5G38850AT3G04220AT5G44940*	Biotic resistance	Increased resistance to *Botrytis cinerea* B1301 strain	[91]
*Arabidopsis*	miR472	*NBS-LRR*	Biotic resistance	Increased resistance to *Pseudomonas syringae* Pv. *tomato* (*Pst)* DC3000	[92]
Cucumber	miR164d miR396b NovelmiR1NovelmiR7	*NAC, APE, 4CL, PAL*	Biotic resistance	Increased resistance to *Corynespora cassiicola*	[93]
Potato	miR397	*IbLACs*	Biotic resistance	Upregulates lignin content that provides physical defence and reduces the accumulation of sweet potato virus disease (SPVD)	[94]
Soybean	miR1510	*GmTNL16*	Biotic resistance	Plant hormone signaling and secondary metabolism interact with *Phytophthora sojae* infection	[95]
Soybean	miR1507a, miR1507c, miR482a, miR168a, miR1515a	Five *NBS-LRR* family genes	Biotic resistance	Increased resistance towards soybean mosaic virus (SMV)	[96]
Tomato	miR482, miR2118b	*NLR*	Biotic resistance	Enhanced resistance toward oomycete and bacterial pathogens	[97]
Tomato	miR482b	*NBS-LRR*	Biotic resistance	Enhancement of tomato resistance to *Phytophthora infestans*	[98]
Tomato	sly-miR1916	*STR-2, UGTs, MYB12*	Biotic resistance	Accumulation of α-tomatine and anthocyanins confers biotic stress tolerance	[85]
Tomato	miR166b	*SlHDZ34/45*	Biotic resistance	Reduced pathogen accumulation in *P. infestans*-infected plants	[99]
Tomato	miR482/2118	*NLR*	Biotic resistance	Improved resistance to oomycete and bacterial pathogen infection	[97]
Citrus	CsmiR399a.1	*CsUBC24*	Flower development	Abnormal floral development, suppression of anther dehiscence, and diminished pollen productivity	[100]
*Arabidopsis*	miR396	*GRF1, GRF2, GRF3, GRF4, GRF7, GRF8, GRF9*	Flower development	Rescues abnormal pistils and siliques	[101]
Cotton	Gh-miR156	*CHLI*	Flower development	Early flowering	[102]
Tomato	miR1917	*SlCTR4*	Fruit development	Increased biomass, longer terminal leaflet, bigger floral organ and better fruit and seed size	[103]
Tomato	Sly-miR164a	*NAC1, NAM3*	Fruit development	The amount of hydrogen peroxide (H_2_O_2_) in the fruit decreased, and its firmness increased in post-harvest chilling	[104]
Tomato	Sly-miR171e	*GRAS24, CBF1, GA2ox1, COR, GA3, GA20ox1, GA3ox1*	Fruit development	Reduced chilling injury (CI) index, lower hydrogen peroxide (H_2_O_2_) level, and greater fruit firmness after harvest	[105]
Kiwi	miR164	*AdNAC6, AdNAC7*	Fruit development	Faster fruit ripening	[106]
Tomato	sly-miR159	*SlGA3ox1, SlGA3ox2*	Fruit morphology	Larger fruit	[107]
*Arabidopsis*	miR169d	*YUC2, PIN1, ARF1, ARF2*	Leaf development	More and larger leaves	[108]
*Arabidopsis*	miR827	*GPLα*	Leaf development	Reduced *PTP1* gene expression, decreased leaf phosphate	[109]
Silver birch	BpmiR164	*BpCUC2*	Leaf development	Reduced internodes and irregular leaf forms	[110]
Rice	miR394	*LC4*	Leaf development	Increased leaf angles	[111]
*Arabidopsis*	miR164	*GhCUC2*	Plant architecture	Decreased length and number of lateral branches	[82]
Cotton	miR164	*GhCUC2*	Plant architecture	Decreased length and number of lateral branches	[82]
Cucurbit	miR159	*GAMYB*	Plant architecture	Dwarf with smaller leaves	[112]
Potato	miR160a/b	*StARF10,* *StARF16*	Plant architecture	Shorter roots, more lateral roots, and less fresh root weight	[80]
Rice	OsmiR166	*OsHox32*	Plant architecture	Increased thickness of cell wall and culm strength	[113]
Rice	miR166	*OsHB4*	Plant architecture	More drought tolerance with rolled-leaf phenotype and reduced xylem vessel diameter	[76]
Rice	miR159	*OsGAMYB, OsGAMYBL1*	Plant architecture	Reduced height and stem diameter, flag leaf length, main panicle, spikelet hulls, and grain size	[53]
Rice	miR528	*DWARF3 (D3)*	Plant architecture	Higher plant height due to lower abscisic acid and higher gibberellin	[114]
Wheat	miR319	*TaPCF8*, *TaGAMYB3*	Plant architecture	Increased plant height, reduced tiller number, spikes and flag leaves, thicker culms, and higher grain output	[115]
Poplar	miR828	*MYB171, MYB011*	Plant architecture	Increased expression of lignin biosynthesis genes and lignin accumulation	[116]
*Arabidopsis*	miR165/166	*HD-ZIP III*	Plant development	Increased auxin content and decreased auxin sensitivity	[117]
Tomato	sly-miR171a-b	*SlHAM*, *SlHAM2*	Plant development	Delayed anther development, increased shoot branching and compound leaf morphogenesis	[118]
Poplar	miR393	*FBL* family members	Plant development	Taller, thicker, more internodes, broader phloem, xylem, and cambium cell layers, higher lignin content	[119]
Soybean	GmSTTM166	*ATHB14-LIKE*	Plant development	Stunted growth	[120]
*Arabidopsis*	AtMIR396a	*bHLH74*	Root development	Longer roots	[121]
*Arabidopsis*	miR397b	*LAC2OX*	Root development	Elevated *LAC2* transcript, decreased lignification in root xylem, and lengthened primary roots	[122]
Rice	miR390	*ARFs*	Root development	Decreased salt tolerance with inhibited lateral root growth	[123]
Tomato	miR858	*SlMYB7*-*like*	Secondary metabolite	Increased anthocyanin biosynthesis gene expression under normal conditions	[124]
Tobacco	nta-miRX27	*QPT2*	Secondary metabolite	Increased nicotine biosynthesis	[125]
Persimmon	DkmiR397	*DkLAC2*	Secondary metabolite	Increased accumulation of proanthocyanins	[126]
Citrus	miR171	*CsSCL*	Somatic embryogenesis (SE)	Weakened somatic embryogenesis (SE) competence	[127]
Rice	miR319	*OsTCP21, OsGAmyb*	Yield	Improved tiller number and grain yield	[128]
Rice	miR398	*Os07g46990*	Yield	Smaller panicles with fewer grains and late flowering	[69]
Rice	miR1432	*OsACOT*	Yield	Increased grain weight by increasing grain filling rate	[129]

## Data Availability

Not applicable.

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
