# Peer review of "Overview of Repressive miRNA Regulation by Short Tandem Target Mimic (STTM): Applications and Impact on Plant Biology"

_plants, 2023, doi:10.3390/plants12030669_

Round 1
Reviewer 1 Report
This article provides an overview of the development history of the STTM approach and its application to inhibit miRNA expression and its effects on plant development, yield, and stress response. It will enhance our more comprehensive and in-depth understanding of STTM applications, but the paper still needs improvement before acceptance for publication, especially in the accuracy of language expression. My detailed comments are as follows.
1. Gene names should be in italics, and the full name should be indicated as “MIRNA” when an abbreviated version of “MIR” appears for the first time on Line 54.
2. The authors should check and revise all sentences carefully to ensure the accuracy of language expression, some examples as the following sentences or words show.
(1) “MIR genes are transcribed by the gene promoters and RNA polymerase II” on Line 77 should be changed to “MIR genes are transcribed from the gene promoters by RNA polymerase II”.
(2) “also known as a primary transcript (pri-miRNAs)” on Line 79 should be changed to “also known as a primary transcripts (pri-miRNAs)”.
(3) “All these proteins process pri-miRNA into pre-miRNA in Dicing bodies (D-bodies)” Lines 86-87 should be changed to “The DCL1, HYL1, and SE proteins can form Dicing complexes and process pri-miRNAs into pre-miRNAs in Dicing bodies (D-bodies)”.
(4) “DCL1 cleaves the pre-miRNA, followed by the methylation at the 3′ end by sRNA methyltransferase Hua Enhancer 1 (HEN1) to protect it from exonuclease” on Lines 93-95 should be changed to “DCL1 cleaves the pre-miRNA, followed by the methylation at the 3′ end by sRNA methyltransferase Hua Enhancer 1 (HEN1) to protect it from the degradation by exonuclease”.
(5) “This miRNA will then combine with Argonaute 1 (AGO1) through the formation of the RNA-induced silencing complex (RISC)” on Lines 93-94 should be changed to “The miRNA duplex will be loaded into Argonaute 1 (AGO1) to form the RNA-induced silencing complex (RISC)”.
(6) The sentence “miRNA will guide the cleavage of the target transcript (mRNA), resulting in mRNA degradation by RISC that cuts the mRNA and ultimately reduce the abundance of mature transcript mRNA for the translation process by the ribosome and induce gene repression.” on Lines 105-107 expressed not very clear and should be reorganized.
(7) “miRNA cleaving activities depend on the opposite of nucleotide 10-11” on Lines 110-112 should be changed to “miRNA-RISC cleaving targets usually depends on the complementarity of the 10-11th nucleotides from 5′ end of miRNA”.
(8) “They can inhibit miRNA by immersion in its chemical, which is more severe if the plant is treated” on Lines 156-157 should be changed to “They can inhibit miRNA function by immersion in its chemical, which is more severe if the plant is treated”.
(9) Do you mean “ (miR156a-j, miR820a-c, miR-169f, g)” on Line 158 is “(miR156a-j, miR820a-c, miR-169f-g)”?
(10) “pre/pri miRNA” on L139 should be “pre/pri-miRNA”.
(11) Do you mean “T5G40910” is “AT5G40910” on Line 8 of Table 1?
(12) “it still lacks understanding of how to plant tolerance toward stress” on Line 327 should be changed to “it still lacks an understanding of how plants tolerate stresses”.
3. The authors need to graphically display the biosynthesis and target regulation of plant miRNAs in the second part “2.miRNA Biogenesis and Its Regulation in Plant Gene Transcript” to help readers better understand these processes.
4. The authors need to graphically compare the methods mentioned in this paper, such as TM, STTM, SP, AMO, and clearly distinguish them in the third section “3.Inhibition Mechanism of Target Mimic That Confers Mature miRNA Repression in Plant”. In addition, please describe in the text why the STTM method is more commonly used, and what advantages it has over FTM, SP, and other methods.
5. The authors need to graphically show the development history of STTM methods in the fourth section “4.Strategies of Short Tandem Target Mimic (STTM) Application in miRNA Loss-Of- Function Studies” and compare the improvements of these methods.
6. Figure resolution needs improvement.
7. Please check the reference format carefully in the MDPI plants journal. Links to Google Scholar, CrossRef not DOI should be added at the end of each reference.
Reviewer 2 Report
The review performs an extensive and comprehensive study of the interesting an relativeley new topic of STTM. References and graphics are very informative, but the text needs extensive editing prior to be ready for acceptance.
Major points: Part of the information is scattered around the text. For instance, STTM is first mentioned in line 69, mentioned again in line 120, 160, 171, 183 and several other parts of the paragraph, but the description and definition of what is a STTM appears in line 201. Is better to describe it in the first mention, Presenting the information in an ordenated manner will improve the readibility, and probably shorten the manuscript.
Also the conclusion must be rewritten, as most of it speaks about miRNA which are not the primary focus of this review, and only the last two sentences are dedicated to STTM. Please, summarize the main points related to STTM and include this in the conclusion.
The english must be thoroughly edited. For instance:
Line 20-21: rewrite the sentence. Difficult to understand.
Line 53-54: there are.
Line 72: "we elaborate development", this does not make sense. Please correct.
Line 124: has been.
Line 137: "The undesirable phenotype". Please explain what is an undesirable phenotype. Less yield? Ugly plants? Prone to pests?
Line 138--> Have been.
Line 174--> activity
Line 417--> "STTM with leaf". Please explain what does this mean.
Round 2
Reviewer 2 Report
After reviewing the revised version in the manuscript I think that the authors hace committed with all the points raised in previous reviews. Therefore I recommend publication